# Comparative Analysis of the Global Warming Potential (GWP) of Structural Stone, Concrete and Steel Construction Materials

Jonathan Kerr [1], Scott Rayburg [1,*], Melissa Neave [2] and John Rodwell [3]

1   Department of Civil and Construction Engineering, Swinburne University of Technology, Hawthorn, VIC 3122, Australia; kerrjp97@gmail.com
2   School of Global, Urban and Social Studies, RMIT University, Melbourne, VIC 3001, Australia; melissa.neave@rmit.edu.au
3   Department of Management & Marketing, Swinburne University of Technology, Hawthorn, VIC 3122, Australia; jrodwell@swin.edu.au
*   Correspondence: srayburg@swin.edu.au; Tel.: +61-(3)-9214-4944

**Abstract:** The manufacturing and construction industries have always been large contributors to global $CO_2$ emissions, largely as a consequence of material choices. Two of the most commonly used building materials are concrete and steel, but both of these industries have been identified as large sources of atmospheric $CO_2$. Therefore, reducing the use of these materials and finding alternatives to them that meet the engineering requirements of a design, while also minimizing emissions, is becoming increasingly important. Stone in its natural form is a zero-carbon emission material and has strong physical properties that make it a viable substitute for concrete and steel, across a range of applications. Yet research into the potential use of stone by the construction industry remains rare. The aim of this research is to investigate whether the use of stone as a building product is a feasible alternative in terms of carbon emissions. This study compares data from 11 Environmental Product Declarations (EPDs) that provide Life Cycle Analysis (LCA) assessments of their considered product (i.e., types of dimensional stone, concrete, or steel). However, this research also highlights some shortcomings in the EPDs that point to a need for greater legitimate engagement with this tool, and for more consistency between the data being presented in EPDs. Global Warming Potential (GWP) data are compared between products to determine the difference in carbon emissions. The results indicate that GWP values for dimensional structural stone (135 kg.$CO_2$/m$^3$) are 45–75% lower than the concrete products considered in this investigation (246–514 kg.$CO_2$/m$^3$), and over 99% lower than certain steel products (22,294–29,202 kg.$CO_2$/m$^3$). This research indicates that stone is demonstrably better in terms of its GWP, and that a more extensive use of structural stone represents a key opportunity for the construction industry to reduce its $CO_2$ emissions.

**Keywords:** sustainable buildings; green buildings; alternative construction materials; global warming potential

## 1. Introduction

Buildings represent an important component of a low-carbon future. According to the Intergovernmental Panel on Climate Change (IPCC), buildings are responsible for a significant share of carbon emissions, accounting for 19% of global energy-related $CO_2$ emissions, a third of black carbon emissions and 51% of global electricity consumption [1,2]. In addition, emissions associated with buildings are expected to significantly increase over the next decade as populations continue to grow and communities in developing nations are given improved access to housing and electricity networks.

Statistics such as these have long made buildings obvious targets for emissions reductions. Indeed, in its first assessment report published in 1990, the IPCC explicitly recognized the potential of the building sector to contribute to reducing global $CO_2$ emissions. Since

then, most work on this has focused on reducing building energy needs by proposing strategies such as: using energy efficient lighting and electrical appliances; placing insulation in walls, floors, and ceilings to limit the need for additional heating/cooling; employing energy efficient alternatives to traditional heating/cooling devices (e.g., heat pumps); orienting buildings to minimize solar exposure; and using vegetation to help shade buildings, among other potential actions. These strategies represent important mechanisms for ensuring that future buildings place less of a burden on the global carbon network. However, an often-neglected component of the impact that buildings have on $CO_2$ emissions is carbon that is released during their construction phase, which can be significant, and is largely a function of the chosen building materials [3].

Currently, the most commonly used building material globally is concrete. Concrete, a hard composite of aggregate (most commonly sand and gravel) and cement, is widely used because it is workable and has an extremely high production rate, meaning it can usually keep up with demand. Innovations that continue to improve the efficiency of concrete use, such as self-climbing forms for high rise construction [4] and precast panels for quicker onsite erection, have also contributed to the ongoing popularity of concrete as a building material. Another common construction material is steel, which, despite having a longer lead (or manufacturing) time than concrete, is extensively and increasingly being used by the construction industry because of its flexibility, strength, and fast onsite erection potential [5].

Unfortunately, both the cement and the iron and steel industries make significant contributions to $CO_2$ emissions. For example, in 2018 cement production emitted an estimated 1.57 billion tonnes of $CO_2$ [6], which represented approximately 4.2% of emissions associated with fossil fuels [7]. Roughly half of these cement-related emissions were released as part of the production of cement clinker [8], which is composed of nodules of limestone and minerals that have been super-heated in a kiln. Clinker production remains a common component of cement but has limited options for reducing emissions associated with its production. Likewise, the production of steel is credited with contributing 7–9% of global $CO_2$ emissions [9,10]. These emissions come from several points along the steel production process, but a significant percentage result from the blast furnaces that are used to transform iron ore into steel [1,11].

The extensive use of concrete and steel as building materials makes the construction industry a significant contributor to global $CO_2$ emissions [12] and points to the need for this industry to consider using sustainable products [13]. Although the construction industry has been slow to adopt alternative building materials, such alternatives have started to become more common, typically involving the inclusion of natural materials into the construction phase. These natural alternative materials include rammed earth, wood (cross-laminate timber, bamboo, fibreboard, etc.), straw, cork, and stone. In addition, some non-conventional materials have also started to appear, such as hand-made alternative building materials usually produced from industrial and agricultural wastes, or other renewable materials [14,15].

Although these alternative building materials work well in residential or small-scale production, only wood and stone are currently considered to have the physical and production capabilities to enable their inclusion into multiple-story structures. As such, over the last decade research has been undertaken to investigate the validity of constructing multi-story timber and timber-hybrid structures and innovations in the structural system, and the application of new engineering wood materials have allowed 10 to 24 story timber hybrid buildings to be constructed in Australia and Austria [16]. However, far less research has been conducted to investigate the viability of using stone to construct low and high-rise structures. Indeed, a comprehensive and highly cited study comparing the embodied energy of various building materials did not even consider stone [17]. This is a serious limitation, as stone itself has a zero-carbon footprint, is highly durable and has immense potential for reuse. Indeed, of all-natural construction materials, stone has proven to be one of the most durable, as demonstrated by its use in some of our largest and oldest historical

constructions (such as the pyramids and ancient buildings in Egypt, South America and Asia, and churches and cathedrals throughout the world). Despite this evidence of stone's utility, little research has considered how a large-scale shift to stone as a preferred building material in the modern era could impact on global $CO_2$ emissions.

In light of the under-appreciation of stone as a sustainable building material, the aim of this research is to determine the potential benefits of using stone compared to concrete and steel in construction, particularly considering whether the use of structural stone could result in a reduction of carbon emissions for the building industry. To achieve this aim, the study has the following objectives:

(1) To produce a comparative estimate of the carbon emissions of structural stone, concrete, and steel;

(2) To predict potential increases/decreases in carbon emissions when using structural stone compared to concrete and steel equivalents;

(3) To assess whether stone provides a viable alternative to building with concrete and steel.

## 2. Materials and Methods

To achieve those objectives, this study uses data on carbon emissions for concrete, steel, and stone that have been published in Environmental Product Declarations (EPD). An EPD is a third-party verified document that provides information about the environmental impact of a particular product [18]. These voluntary devices are being increasingly used as tools for environmental assessment [19] and are publicly available.

To compare the carbon emissions for concrete, steel, and stone, 11 EPDs were investigated (Table 1). Data within these EPDs are presented in the form of a Life Cycle Analysis (LCA), which is a method that provides a 'cradle-to-grave' analysis of the environmental impacts of manufactured products. LCAs are widely used to calculate environmental costs and benefits and, for the purpose of this study, the results of the LCAs for different building products are used to compare the volumes of carbon dioxide emitted by each. The building materials considered in this study include concrete (with a strength of 30 MPa and higher), steel, and dimensional stone, which refers to stone that has been quarried, cut, and shaped specifically for construction purposes.

**Table 1.** Links to the 11 Environmental Product Declarations (EPD) examined in this study, grouped according to product type.

| **Dimensional Stone** |
|---|
| Levantina y asociados de minerals<br>Marble and limestone slabs—Used for aesthetic products<br>https://www.aenor.com/Producto_DAP_pdf/GlobalEPD_EN15804_001_ENG.pdf (accessed on 5 October 2020) |
| Minera Skifer<br>Natural stone quartzite schist—Used for structural products<br>https://mineraskifer.no/wp-content/uploads/2015/04/EPD-Oppdal-quartzite-natural-cleft-surface-broken-or-sawn-edge.pdf (accessed on 5 October 2020) |
| **Concrete** |
| Allied Concrete<br>Ready-mix concrete<br>https://www.alliedconcrete.co.nz/assets/technical-resources/files/Sustainability/Allied-Concrete-Environment-Product-Declaration-2019.pdf (accessed on 8 October 2020) |
| Holcim (Australia)<br>ViroDecs$^{TM}$ ready-mix concrete<br>https://epd-australasia.com/wp-content/uploads/2019/07/Holcim-EPD-ViroDecs_v1.1.pdf (accessed on 8 October 2020) |

**Table 1.** *Cont.*

| |
|---|
| Institut Bauen and Umwelt e.V.<br>Precast concrete ground beams<br>https://www.britishprecast.org/Sustainability/EPDs/BPAS-1m3-Concrete-Ground-Beam.aspx<br>(accessed on 5 October 2020)<br>Generic ready-mix concrete<br>https://www.concretecentre.com/TCC/media/TCCMediaLibrary/PDF%20attachments/<br>Generic-ready-mixed-concrete.pdf (accessed on 5 October 2020)<br>Agilia™ ready-mix concrete<br>https://www.aggregate.com/sites/aiuk/files/atoms/files/agilia_ready-mixed_concrete_epd_<br>final.pdf (accessed on 5 October 2020) |
| Votorantim Cimentos<br>Concrete<br>https://api.environdec.com/api/v1/EPDLibrary/Files/26d3e953-3198-4fea-b97b-6633ce982b9<br>b/Data (accessed on 8 October 2020) |
| **Steel** |
| BlueScope<br>Steel: Welded beams and columns<br>https://cdn.dcs.bluescope.com.au/download/environmental-product-declaration-welded-<br>beams-and-columns (accessed on 4 October 2020) |
| InfraBuild Australia<br>Steel: Hot rolled structural section and merchant bar products<br>https://www.infrabuild.com/wp-content/uploads/sites/8/2020/10/5-EPD_IB-Steel-Centre-<br>Hot-Rolled-Structural_2022.pdf (accessed on 9 October 2020) |
| Liberty Primary Steel<br>Steel: Hot rolled structural and Rails<br>https://www.libertygfg.com/media/334834/2-epd_liberty-hot-rolled-structural-rail.pdf<br>(accessed on 9 October 2020) |

In the construction industry, stone can be used for a myriad of purposes, but these can be crudely subdivided into aesthetic and structural sub-groups. Aesthetic stone refers to materials that are used for ornamental purposes, such as indoor and outdoor flooring and surface veneers (Figure 1). Structural stone refers to materials that are explicitly used for structural purposes, such as pillars or building blocks for foundations, walls, stairs, or floors (Figure 2). Because aesthetic products are being used explicitly for their visual appeal, they are treated (e.g., polished and edged) more intensely than structural stone materials, and may have higher carbon emissions than their structural equivalents. Consequently, the dimensional stone products considered in this study include products from these two sub-groupings. Unfortunately, because dimensional stone is not currently widely used for structural purposes, EPDs for these products were limited and therefore only one was included in this study.

The LCAs presented in these EPDs that were used in this research were structured in accordance with ISO 21930:2017, the standard for sustainability in buildings that is summarized in Table 2. For each of the materials under investigation, the following four life-cycle stages were considered: Production (A1–A3); Construction and installation (A4–A5); Use (B1–B7); End of life (C1–C4). This study particularly focuses on emissions resulting from the Production Stage (A1–A3) because it has the largest potential for revealing variations between material types. The other stages are also considered but are less likely to exhibit substantial differences with material type as they use many of the same steps irrespective of the chosen building material, and also because the EPDs often provided less information about these stages.

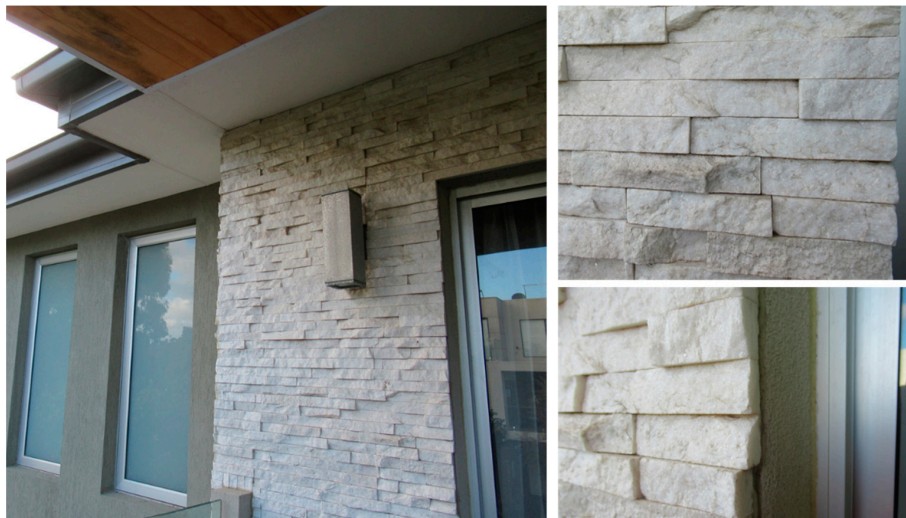

**Figure 1.** Example of stone used for aesthetic purposes. This veneer is composed of small pieces of white quartz that have been glued together to create tiles for cladding. These tiles are ~12 mm thick.

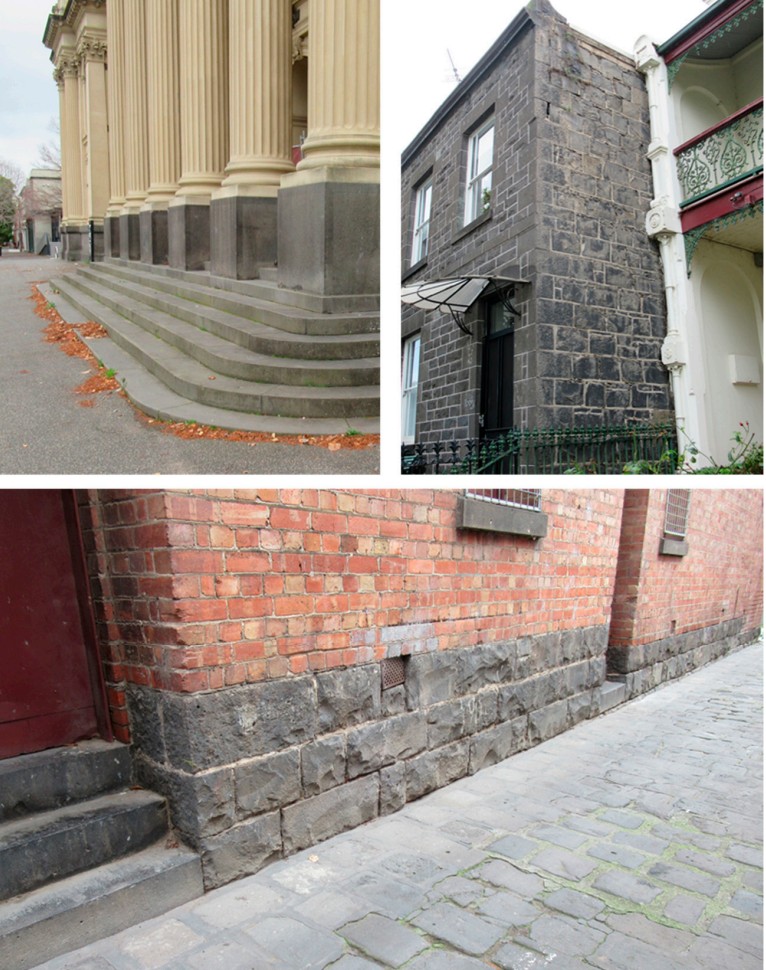

**Figure 2.** Examples of stone being used for structural purposes, including for building foundations and walls, stairs, window sills, and roads. The stone here is 'bluestone', a type of basalt from Victoria, Australia.

**Table 2.** Life cycle stages for building materials (ISO 21930:2017).

| Production | | | Construction | | Use | | | | | | | | End of Life | | |
|---|---|---|---|---|---|---|---|---|---|---|---|---|---|---|---|
| Extraction | Transport | Manufacturing | Transport to Site | Installation | Use | Maintenance | Repair | Replacement | Refurbishment | Operational Energy Use | Operational Water Use | Deconstruction/ Demolition | Transport | Waste | Disposal |
| A1 | A2 | A3 | A4 | A5 | B1 | B2 | B3 | B4 | B5 | B6 | B7 | C1 | C2 | C3 | C4 |

Once collected, the EPD data were compared to estimate the overall embodied emissions associated with using different building materials, expressed in terms of each product's Global Warming Potential (GWP). The GWP is an equivalent measure of the $CO_2$ emissions associated with production and in this study is presented as kilograms of carbon dioxide per volume/weight of material produced. Within the 11 study EPDs the functional units for GWP were presented as either kg, t or $m^3$. Thus, comparisons between products required an initial conversion of some of the data so that all GWP units were in $m^3$. To convert GWP units of $kg.CO_2/t$ to $kg.CO_2/m^3$ the original value was multiplied by the density of the product, which was also obtained from the EPD. To convert GWP units of $kg.CO_2/kg$ to of $kg.CO_2/t$ the original value was multiplied by 1000. This resulted in all products being presented in comparable units and enables a preliminary assessment of the environmental costs resulting from using different products. These findings can then be extrapolated to predict the implications for the construction industry more generally if a large-scale shift in building materials occurs.

Most of the EPDs reported individual GWP values for the product under consideration. However, the values for ready-mix concrete, using Holcim cement provided by Allied Concrete, were given for multiple batching plants across New Zealand, categorized according to strength. For the purpose of this study, these GWP values were averaged to provide a single estimate of the carbon emissions associated with each strength, and the average data have been presented with an indication of the number of batching plants used to achieve this value.

## 3. Product Emission Comparisons

Carbon emissions in the form of Global Warming Potential (GWP) for the building materials of structural stone, concrete, and steel obtained from the 11 EPDs investigated in this study are presented and discussed according to the four life-cycle stages defined by ISO 21930:2017. These data allow for a preliminary comparison of the carbon emissions associated with each product.

### 3.1. Production Stage (A1–A3)

The Production Stage of the LCA considers the costs of extracting, transporting, and manufacturing a product. Dimensional stone is typically extracted from a quarry as a large block that, depending on the original size, is then subdivided (cut down) into smaller blocks (A1). The smaller stone blocks are transported (A2) out of the quarry to a processing factory (or mill), where they may be cut down further and/or finished (A3). Thus, because the source material naturally presents in a mostly usable form there are relatively few steps required to produce dimensional stone for building purposes. In contrast, the production of concrete and steel is more complicated. For example, concrete requires the extraction of multiple source materials (e.g., limestone, slate, clay, and/or gypsum) from different locations (A1). These materials are each transported to a central processing facility (A2) where they are then crushed and blended, before being burned in a kiln and ground to produce the final product (A3). Likewise, steel is made from coal, iron ore, and limestone that are extracted from multiple locations (A1) and brought to a central processing site (A2). The coal is initially burned under high heat to produce coke, which is then mixed with iron ore and limestone in a furnace to produce steel (A3). Table 3 provides the GWP data for the LCA Production Stage (A1–A3) obtained from the 11 EPDs investigated in this study.

**Table 3.** Global Warming Potential for the Production Stage (A1–A3) of the LCA for various dimensional stone, concrete, and steel products.

| Product | Source * | Density # (t/m$^3$) | GWP # (kg.CO$_2$/t) | GWP (kg.CO$_2$/m$^3$) |
|---|---|---|---|---|
| **Dimensional Stone** | | | | |
| *Aesthetic* | | | | |
| Marble slabs–Crema Marfil Coto | 1 | 2.67 | 60.7 | 162 |
| Marble slabs–Marron Emperador | 1 | 2.67 | 99.1 | 265 |
| Limestone slabs–Caliza Capric | 1 | 2.67 | 90.6 | 242 |
| Quartzite schist–even thickness | 2 | 2.74 | 93.3 | 272 |
| *Structural* | | | | |
| Quartzite schist–natural cleft surface | 2 | 2.74 | 49.3 | 135 |
| **Concrete** | | | | |
| *Strength = 30 MPa* | | | | |
| Concrete FCK 30 MPA BR.1 10 ± 2 | 8 | | | 267 |
| Ready-mix using Holcim cement | 3 | (44 batching plants) | | 319 |
| Generic ready-mix | 5 | | | 246 |
| *Strength = 32 MPa* | | | | |
| General mix | 4 | | | 343 |
| *Strength = 35 MPa* | | | | |
| Ready-mix using Holcim cement | 3 | (43 batching plants) | | 351 |
| *Strength = 40 MPa* | | | | |
| Agilia$^{TM}$ Ready-mix Concrete | 6 | | | 373 |
| Ready-mix using Holcim cement | 3 | (42 batching plants) | | 386 |
| Precast ground beam (100 kg of steel reinforcement) | 7 | | | 447 |
| *Strength = 45 MPa* | | | | |
| Ready-mix using Holcim cement | 3 | (18 batching plants) | | 411 |
| General mix | 4 | | | 405 |
| *Strength = 50 MPa* | | | | |
| Ready-mix using Holcim cement (Normal) | 3 | (33 batching plants) | | 444 |
| Ready-mix using Holcim cement (Special) | 3 | (2 batching plants) | | 358 |
| General mix | 4 | | | 514 |
| **Steel** | | | | |
| Structural welded beams and columns | 9 | 7.85 | 2840 | 22,294 |
| Hot rolled structural section | 10 | 7.85 | 3720 | 29,202 |
| Merchant bar | 10 | 7.85 | 1520 | 11,932 |
| Hot rolled structural section | 11 | 7.85 | 3320 | 26,062 |

* Data are sourced from the EPDs listed in Table 1. # GWPs provided per tonne were multiplied by density to convert to m$^3$.

The greater complexity of, and the use of heat in, the production processes of concrete and steel mean those products generally have larger CO$_2$ emissions than dimensional stone for the Production Stage, although the extent of that difference varies depending on the type of dimensional stone, concrete, or steel being considered. The GWP examples for dimensional stone provided in Table 3 are 135 kg.CO$_2$/m$^3$ for the structural product, and range from 162 to 272 kg.CO$_2$/m$^3$ for the aesthetic products. Interestingly, the extreme GWP values for the total range of dimensional stone materials considered (135–272 kg.CO$_2$/m$^3$) are actually for the same extracted product of quartzite schist. The wide range indicates the importance of variations in product treatment at the mill (A3) when it comes to calculating GWP. The GWP values for the aesthetic marble and limestone slabs vary from 162 to 265 kg.CO$_2$/m$^3$. Therefore, the calculated GWP for natural stone will depend on what type of stone is being considered (in this case, marble, limestone, or quartzite schist) and how the stone is subsequently treated (whether it is being used for aesthetic purposes), and there is likely to be considerable variation between these. In addition, the location of a quarry site

in relation to its manufacturing plant (or mill) will also play a large role in determining the associated carbon emissions for the Production Stage of the LCA (A–A3). For example, the limestone slabs considered in this study had an average GWP of 242 kg.$CO_2$/m$^3$ but nearly one-third of these emissions came from stage A2, or transportation to the manufacturing site. In contrast, <10% of the emissions for the two marble products were associated with stage A2. The weight of stone means that increased distances between a quarry and its manufacturing site are going to have an immediate impact on the GWP for a dimensional stone product. Despite these issues, natural stone has fewer production demands than other materials because the stone itself exists naturally in the landscape. The fewer production demands help to reduce the overall GWP of stone.

The GWP values for the concrete products presented in Table 3 range from 246 to 514 kg.$CO_2$/m$^3$. These results indicate that the GWP for concrete can also fluctuate significantly as its production involves various processes and different original content materials, leading to a variety of product strengths. Concrete grades can range from 17.5 MPa to ~50 MPa, with higher grades emitting more $CO_2$ due to the adjusted contents and extended processing requirements. For example, a comparison of the 32 MPa general concrete mix to the 50 MPa equivalent considered in this study, is associated with an approximately 50% increase in GWP (Table 3). In addition, concrete that is used for structural purposes often also includes steel reinforcement for extra support and strength. Although this might represent a relatively small percentage of the total weight of the final product, the inclusion of even a small percentage of steel increases the total GWP. The data in Table 3 demonstrate that an extra 100 kg of steel reinforcement in a 1 tonne concrete mix results in a GWP increase of 60–70 kg.$CO_2$/m$^3$ over plain concrete.

Finally, the results of this study indicate that the extensive use of consumables and heat required for steel manufacturing means that steel has extremely high GWPs. Indeed, the GWP values for steel presented in Table 3 range from 11,932 to 29,202 kg.$CO_2$/m$^3$, with the hot-rolled structural products recording the highest GWPs. Steel is a widely used construction material and so these findings indicate it has an important role to play in the total carbon emissions that are attributed to the construction industry, associated with the Production Stage of the LCA. Indeed, using dimensional stone, instead of structural welded or hot-rolled steel products, could reduce $CO_2$ emissions by over 99%.

Collectively, the data in Table 3 reveal that there is capacity for the construction industry to reduce its carbon emissions by selecting products with a lower GWP, and that dimensional stone can make a substantial contribution to this. For example, converting from either a ready-mix or reinforced 40 MPa concrete product, to a structural dimensional stone product, can result in a 64–70% reduction in $CO_2$ emissions based on considerations of the Production Stage of the LCA.

*3.2. Construction Stage (A4–A5)*

The Construction Stage of the LCA (Table 2) considers carbon emissions associated with the transportation of a product from its manufacturing site to the construction site, and the installation of the product at that site. The carbon emissions for both components vary depending on the nature of individual projects. For transportation from the gate to the site (A4), the major factor is the required travel distance and Table 4 provides estimates for that taken from the study EPDs. Unfortunately, only three of the 11 EPDs considered in this study had data for the Construction Stage (A4–A5), indicating that even when an EPD is produced it does not necessarily contain all of the information required to complete an LCA. In addition, A4 is not calculated using a standard transportation distance, giving vastly different results for this stage. To correct for this, the GWP for A4 for each of the four examples was calculated over a standard distance of 50 km. The results of these analyses are that the GWP per km is relatively similar for the three products of quartzite schist, generic ready-mix concrete, and steel-reinforced concrete.

**Table 4.** Global Warming Potential for the Construction Stage (A4–A5) of the LCA for various dimensional stone and concrete products. Note: No equivalent data were available for steel.

| Product | Source * | A4 EDP Distance (km) | A4 EDP GWP # (kg.CO$_2$/m$^3$) | A4 GWP# over 50 km (kg.CO$_2$/m$^3$) | A5 GWP # (kg.CO$_2$/m$^3$) |
|---|---|---|---|---|---|
| **Dimensional Stone** | | | | | |
| *Aesthetic* | | | | | |
| Quartzite schist–even thickness | 2 | 650 km | 119.7 | 9.2 | 108.8 |
| *Structural* | | | | | |
| Quartzite schist–natural cleft surface | 2 | 650 km | 119.7 | 9.2 | 62.7 |
| **Concrete** | | | | | |
| *Strength = 30 MPa* | | | | | |
| Generic ready-mix | 5 | 12 km | 2.01 | 8.4 | 0.19 |
| *Strength = 40 MPa* | | | | | |
| Precast ground beam (100 kg of steel reinforcement) | 7 | 182 km | 29.7 | 8.2 | 0.11 |

* Density and GWP data are sourced from the 11 EPDs listed in Table 1. # GWPs provided per tonne were multiplied by density to convert to m$^3$.

Comparisons between the declared values for A5 were also complicated by differences in the level of detail considered for the installation phase of the EPDs. Thus, dimensional stone, which specified that mortar, water and electricity consumption, waste treatment and a 10% material loss had been considered in its calculation, had considerably higher GWPs for installation (A5) than either of the two concrete products. Furthermore, the aesthetic stone product had a higher GWP value for A5 than the structural product, indicating the need for more materials in the installation of the aesthetic stone. In contrast, the EPDs for both the generic ready-mix and steel-reinforced concrete discussed only material loss in association with their calculation of A5, and these were presented as 3% and 0.009%, respectively. None of the EPDs for the steel products contained data for the Construction Stage (A4–A5) of the LCA.

The lack of data within the EPDs for stages A4–A5, and the lack of consistency between the data that were presented, make it challenging to draw meaningful conclusions about the various contributions of the compared building products when it comes to the Construction Stage. In reality, emissions associated with installation (A5) are difficult to average because they are dependent on the conditions in place for each individual project, such as building type, floor level, machinery availability, and installation method. Consequently, although transport emissions can be averaged for a given distance (A4), the calculation of GWP for an installation (A5) is always going to be challenging and will be inaccurate unless specific details about individual projects are considered—a task that can not be achieved within an EPD.

### 3.3. Use Stage (B1–B7)

Similar to the Construction Stage, the EPDs considered in this study provided limited information on GWPs associated with the Use Stage of the LCA (Table 2), which in addition to the use itself considers maintenance, repair, replacement, refurbishment, and operational energy and water use (B1–B7). Of the 11 EPDs investigated in this study, only three discussed carbon emissions associated with product use, and all three indicated zero or negative GWP values for this stage (Table 5). Although at first glance these results might seem uninteresting, they are actually important because they indicate that the building products being considered in this study are expected to be essentially static for the life of the building itself. In a structural sense, these materials cannot be removed or replaced without major construction works, which would be considered within the Construction Stage of the LCA. In addition, although two of the three EPDs reported zero GWP values for the Use Stage, the generic ready-mix concrete EPD provided a negative GWP. This negative value reflects the occurrence of concrete carbonation, which affects concrete products unless they

are treated. This process results in the sequestration of $CO_2$ and can weaken the impacted concrete product. Collectively, the results for B1–B7 indicate that the building materials of dimensional stone, concrete, and steel are comparable in terms of emissions associated with their use.

**Table 5.** Global Warming Potential for the Use Stage (B1–B7) of the LCA for various dimensional stone and concrete products. Note: No equivalent data were available for steel.

| Product | Source * | GWP # ($kg.CO_2/m^3$) |
|---|---|---|
| **Dimensional Stone** | | |
| *Aesthetic* | | |
| Quartzite schist–natural cleft surface | 2 | 0 |
| *Structural* | | |
| Quartzite schist–even thickness | 2 | 0 |
| **Concrete** | | |
| *Strength = 30 MPa* | | |
| Generic ready-mix | 5 | −19.90 |
| *Strength = 40 MPa* | | |
| Precast ground beam (100 kg of steel reinforcement) | 7 | 0 |

* Density and GWP data are sourced from the 11 EPDs listed in Table 1. # GWPs provided per tonne were multiplied by density to convert to $m^3$.

### 3.4. End of Life Stage (C1–C4) & Recycling Potential

The End of Life Stage of the LCA encompasses emissions association with the deconstruction/demolition (C1), transport (C2), waste processing (C3), and disposal (C4) of a product after its use (Table 2). Of the 11 EPDs considered in this study, eight provided GWP estimates for these categories and included calculations for dimensional stone, concrete, and steel (Table 6). Emissions of $CO_2$ associated with the end-of-life for dimensional stone (both aesthetic and structural) were estimated to be approximately 30 $kg.CO_2/m^3$. Roughly three-quarters of these emissions were a result of an estimated 50 km transportation of the demolished material to a landfill (C2), with most of the remainder coming from disposal (C4).

**Table 6.** Global Warming Potential for the End of Life Stage (C1-C4) of the LCA for various dimensional stone, concrete, and steel products.

| Product | Source * | Density # ($t/m^3$) | GWP # ($kg.CO_2/t$) | GWP ($kg.CO_2/m^3$) |
|---|---|---|---|---|
| **Dimensional Stone** | | | | |
| *Aesthetic* | | | | |
| Quartzite schist–even thickness | 2 | 2.74 | 11.1 | 30.41 |
| *Structural* | | | | |
| Quartzite schist–natural cleft surface | 2 | 2.74 | 11.08 | 30.37 |
| **Concrete** | | | | |
| *Strength = 30 MPa* | | | | |
| Generic ready-mix | 5 | | | −9.49 |
| *Strength = 40 MPa* | | | | |
| Precast ground beam (100 kg of steel reinforcement) | 7 | | | −26.67 |
| **Steel** | | | | |
| Structural welded beams and columns | 9 | 7.85 | 56.18 | 441.00 |
| Hot rolled structural section | 10 | 7.85 | 6.92 | 54.32 |
| Merchant bar | 10 | 7.85 | 6.92 | 54.32 |
| Hot rolled structural section | 11 | 7.85 | 6.92 | 54.32 |

* Density and GWP data are sourced from the 11 EPDs listed in Table 1. # GWPs provided per tonne were multiplied by density to convert to $m^3$.

The EPD calculations for dimensional stone did not consider the reuse/recovery/recycling potential of these products, even though there is immense capacity for their reuse. Indeed, one of the most obvious benefits of using stone is its longevity and the ability to either reuse it in its original form, or repurpose it into something else. That is, stone from buildings can be reused in other buildings or redirected into products such as veneers, pavers, garden structures, counter tops, or even crushed rock, with little to no treatment of the original product.

In contrast to those for dimensional stone, the EPDs for concrete calculate the GWP associated with end-of-life for the generic ready-mix and steel-reinforced concrete to be $-9.49$ and $-26.67$ kg.$CO_2$/m$^3$, respectively (Table 6). These low values are based on the assumption that 90% of the concrete product is reused/recovered/recycled while only 10% is sent to landfill. That assumption results in a strong negative value for C3, which likely reflects the avoidance of needing to use natural aggregate. Although concrete can be and is recycled, it is somewhat unrealistic to assume that 90% of the product is being kept out of landfill, which calls the estimate for these GWP values into question.

The EPDs for steel (Table 6) also consider product reuse/recovery/recycling, although the extent of such reuse differs between them. For structural beams and columns, the reuse/recovery/recycling rate was presumed to be 17.4%, which resulted in a GWP of 369.7 kg.$CO_2$/m$^3$ for the waste processing stage (C3) of that product. However, for the hot rolled structural sections, and merchant bar products, the recycling/recovery rate was set at 90%, which resulted in a GWP of 18.92 $CO_2$/m$^3$ for C3 for these products. Once again, this indicates the importance of accurately representing reuse/recovery/recycling rates in the LCA calculations. For completeness, these rates need to represent an actual value, rather than an aspirational one.

## 4. Discussion

Comparisons of the GWP results included in the 11 EPDs considered in this study reveal potential for the construction industry to reduce its high rates of carbon emissions through strategic building material choices, and dimensional stone offers an important alternative resource to consider. Adding up the GWP values for structural dimensional stone, and comparing them to those for either the ready-mix or reinforced 40 MPa concrete products, reveals a potential reduction in embedded carbon of between 35–45%. That is, dimensional stone, particularly products designed for structural purposes, offer a feasible alternative construction material. However, considerations of carbon emissions associated with building materials represent only one reason as to why a particular product might be selected for construction, and means dimensional stone might not be appropriate for all circumstances. A variety of other reasons influence material choices, but an obvious issue to consider is physical capability.

### 4.1. Physical Capabilities

Stone is one of the strongest natural materials and its structural performance provides merit for its use in construction. For example, limestone has a density of 2.711 t/m$^3$ while its compressive strength is 115 N/mm$^2$ or 11,726.74 t/m$^2$. This strength means the pillar height that can be reached before the base block collapses is 4.3 km [20]. Such structural performances indicate the potential value of considering stone for building. However, although stone has good compressive strength, its tensile strength is weaker than that of other products, which limits its use for products such as beams. In particular, concrete reinforced with steel or steel alone are both better alternatives in most tensile situations. This means that while stone can be championed as a viable product for many construction purposes, there are some limits to its potential use.

### 4.2. Data Quality and EPDs

This study examined GWP data from 11 EPDs. An EPD is a document specifically designed to provide quantifiable environmental data that enables comparisons between

products. These documents are used to help those working in the construction industry (such as engineers or architects) make more sustainable product choices. However, this study has revealed some issues with the available EPDs that reduce their potential to inform choices. All of the 11 EPDs considered in this study provided data for the Production Stage (A1–A3) of the LCA; however, only one EPD provided those data for each individual component of this stage. In other words, most EPDs provided a single GWP for the entire Production Stage (A1–A3 combined), which makes it challenging to identify how components such as transportation (A2) and manufacturing (A3) individually contribute to the overall GWP. Yet, where these data were included, they show considerable variation between products, which indicates that understanding the contributions these make to GWP is important.

In addition, only three EPDs considered the Construction Stage (A4–A5). The reality is that the Installation Stage (A5) is difficult to provide averages for because specific circumstances are going to vary for each individual project, and it would be very difficult to represent all of the individual components of the Construction Stage in an EPD. However, transportation to the installation site (A4) could easily be represented in a meaningful manner, and because the emissions associated with transportation may vary depending on product type, it would be beneficial to have access to that information. Likewise, only three EPDs provided estimates for the Use Stage (B–B7) of these products, although that may not be particularly problematic because there are no ongoing carbon emissions associated with this stage for any of the products examined here. The End of Life Stage (C1–C4), however, is important when considering the sustainability credentials of the products used in construction. Of the 11 EPDs considered in this study, only six provided details on the Use Stage, and two of those considered only C3 and C4. What happens to a product at the end of its Use Stage is an incredibly important component of its sustainability credentials and should be included more comprehensively in the EPDs.

In addition to issues with LCA stages not being considered at all within the EPDs, there were also problems with the consistency of some of the data that were provided. The GWP values were calculated for different declared units. For example, one EDP had a declared unit of 1 kg, four EPDs had a declared unit of 1 t and six EPDs had a declared unit of 1 m$^3$ (the concrete products). These differences mean that the published GWP values cannot be immediately compared without some form of data manipulation, something that might be easily missed by a reader/user. Thus, although it was possible to immediately consider differences between concrete products, identifying alternatives to concrete, such as dimensional stone, required the declared units to be transformed from tonnes to cubic metres. There were also inconsistencies for calculations of GWP associated with transportation. The transportation of the original extracted product to the manufacturing site (A2) can be included in the GWP calculation for the Production Stage. But transportation of the product to the installation site (A4) and transportation associated with the end-of-life of the product (C2) will vary depending on the location of the construction site. Therefore, this value should be presented according to a standard that considers the same vehicle type (e.g., 50 t EURO5 truck) travelling the same distance (e.g., 50 km). This would enable direct comparisons between products irrespective of the actual circumstances surrounding A4 or C2. Given that EPDs are designed to facilitate more strategic decision making around sustainability, these issues reduce their validity and undermine their purpose.

## 5. Conclusions

This study investigates the potential benefits of using dimensional stone in construction rather than concrete and steel, particularly considering carbon emissions. The findings indicate that a shift in construction material would result in a major decrease in CO$_2$ emissions. Most of these reductions are associated with the extraction of raw materials and the manufacturing required to produce the products. Stone occurs as a ready to use product, where the energy required to quarry, process, and transport the stone generate its CO$_2$

emissions. In contrast, concrete and steel are made of multiple source materials that need to be transported to a central manufacturing facility, and they both require the input of heat to produce a usable product.

All three of the construction materials considered in this study have positive and negative characteristics, and the final material choice will therefore be dependent on each specific project and its requirements. Steel is very strong and versatile and is widely used for its tensile strength. But steel production generates significant carbon emissions, substantially more than either concrete or dimensional stone. Concrete is a strong and workable material with many different varieties and lower embodied $CO_2$ emissions than steel. However, using dimensional structural stone instead of concrete can further lower GWP values. The data from this study estimate that GWP values for structural stone (135 kg.$CO_2$/m$^3$) are 45–75% lower than the concrete products (246–514 kg.$CO_2$/m$^3$) and over 99% lower than the steel products (22,294–29,202 kg.$CO_2$/m$^3$) considered in this study. However, assessments of the EPDs have revealed two key weaknesses in their application. First, there are very few EPDs available for dimensional stone, which limits our capacity to fully appreciate whether these products are viable alternatives in terms of carbon emissions, and may hinder their consideration by industry professionals. Second, the data within the EPDs were not consistent between products: presenting different LCA stages; aggregating stages; and using different units of measurement and declared units. This makes it challenging to compare products using EPDs, especially for time-poor industry professionals, despite that being their purpose.

Despite the opportunity for substantial GWP reductions, dimensional stone is currently being used for mostly aesthetic reasons, such as veneers on walls. This research provides justification for considering a more extensive use of dimensional stone by the construction industry on the grounds that doing so will help lower carbon emissions, to provide a stronger and more sustainable industry in the future. To help promote this transition, ongoing work is required to produce new EPDs for different products, standardize the EPD approach, and develop industry-ready tools that allow for rapid assessments of the value of structural stone as an alternative to concrete and steel for construction.

**Author Contributions:** Conceptualization, J.K. and S.R.; methodology, J.K. and S.R.; formal analysis, J.K. and S.R.; investigation, J.K. and S.R.; writing—original draft preparation, J.K., S.R. and M.N.; writing—review and editing, J.K., S.R., M.N. and J.R.; supervision, S.R. All authors have read and agreed to the published version of the manuscript.

**Funding:** This research received no external funding.

**Conflicts of Interest:** The authors declare no conflict of interest.

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
