# Peer review of "Comparative Analysis of the Global Warming Potential (GWP) of Structural Stone, Concrete and Steel Construction Materials"

_sustainability, doi:10.3390/su14159019_

Round 1
Reviewer 2 Report
Structural Stone: A Sustainable Alternative to Concrete?
Manuscript Number:
In the present paper authors provide an overview of structural stone and highlights some shortcomings in the Environmental Product Declaration. However, the paper requires some minor improvement before it can be recommended for publication, it is proposed to re-submit a thoroughly revised version of the manuscript, considering the following comments.
- Title and abstract are ok
- Overall recommendation should be reported in one sentence at the end of the abstract
- The authors should overview the recent progress made in the relevant area in the past two years or so.
- Emphasizing the importance of research in introduction
- Please privide some building examples made from stone from diffrent regions in the world such as middle Esat..Yemen, Egypt .. , west .....etc. 5. The paper is well written and it is easy to follow, only the authors needs to go thoroughly revised version to correct the typo-mistake.
- Author should highlight the assumptions and limitations and future research direction of the study.
Reviewer 3 Report
Title: It does not reflect what was actually done. It should be reframed e.g.
i. Structural Stone: A sustainable alternative to concrete with respect to carbon emission OR
ii. Comparative Analysis of the Global Warming Potential (GWP) of Structural Stone, Concrete and Steel Materials
Abstract: State actual values of GWP for stone, concrete and steel products in a range
Keywords: Delete embedded carbon. Add GWP
Introduction: - 2nd line: State Full meaning of IPCC when mentioned for the 1st time.
Table 1: Why provide links? This doesn't provide full information on the paper when reading off line.
Provide summarized data on EPD for the products.
Results and Discussion: A separate section should be provided for this before the sub-section mentioned.
Findings from the study should be compared with results from previous studied.
Reviewer 4 Report
Good paper and very interesting topic. Just a remark on the opportunity of citing some historical example on the use of this "Structural Stone".
Finally, bibliography can be extended and updated.
Author Response
N/A